# Characteristics, Management and Outcomes of Spondylodiscitis in Children: A Systematic Review

**DOI:** 10.3390/antibiotics10010030

**Published:** 2020-12-31

**Authors:** Irene Ferri, Gabriele Ristori, Catiuscia Lisi, Luisa Galli, Elena Chiappini

**Affiliations:** 1Paediatric Infectious Disease Unit, Meyer Children’s University Hospital, Department of Health Sciences, University of Florence, 50139 Florence, Italy; irene.ferri@stud.unifi.it (I.F.); katiuscia.lisi@gmail.com (C.L.); luisa.galli@unifi.it (L.G.); 2Paediatric Orthopaedic and Traumatology Department, Meyer Children’s University Hospital, Department of Health Sciences, University of Florence, 50139 Florence, Italy; gabriele.ristori@meyer.it

**Keywords:** spondylodiscitis, discitis, vertebral osteomyelitis, children, pediatric, antibiotic therapy

## Abstract

Spondylodiscitis (SD) is the concurrent infection of the intervertebral disc and the adjacent vertebral bodies. Currently, there is a substantial lack of structured reviews about this topic. The aim of this study was to systematically review the available literature in order to determine the main features of pediatric SD. A systematic search of MEDLINE database was performed, according to the PRISMA guideline recommendations. Clinical features, laboratory data, radiological signs, treatments strategies, and outcomes were summarized. Studies’ quality assessments were performed using the JBI Critical Appraisal Checklists. A total of 35 retrospective studies were analyzed and 340 children were identified. The most frequently affected age class was 0.5–4 years. The most affected site was the lumbar spine. The most commonly reported symptoms were back pain (37.97%) and refusal to walk/to stand/to sit (49.79%). The most frequently identified pathogen was *Staphylococcus aureus* (n = 33). The most used antibiotics were third generation cephalosporins. The intravenous therapy duration range was 1–25 weeks, the oral therapy duration range was 5 days–36 months. Surgery was used in 5.88% of children. In 29 cases clinical sequelae were documented. This study provides the main features of pediatric SD; it also emphasizes the significant gaps in the literature regarding this topic.

## 1. Introduction

Spondylodiscitis (SD) is a term frequently used to describe a continuum of spinal infections, from discitis to vertebral osteomyelitis through SD, with occasional associated soft-tissue abscess [1,2,3]. The term SD covers vertebral osteomyelitis and discitis. The vascular supply of the vertebral body is important in understanding the typical spread of the infection and in distinguishing the two main patterns of the disease. Historically, discitis and vertebral osteomyelitis were considered two different entities [4]. In discitis, the infection is restricted to the intervertebral disc space, which may lead to disc erosion, whereas in vertebral osteomyelitis the adjacent endplates are affected. However, in more recent studies, children rarely presented with pure discitis. Experts agree that they are two aspects of the same spectrum, which appear in two different stages of the pathological process [5]. In children, the pure discitis reported previously does not occur, due to reasons of vascular anatomy. The network of arterial anastomoses extends into the intervertebral disc; therefore, the infection primarily starts in the disc and then spreads to the vertebral endplates.

The age at the presentation more frequently ranges from two to eight years, but it can affect older children and teenagers too.

Childhood SD is a rare disease, but it can cause significant clinical problems [3]. The infection may expand and may also involve the epidural space, posterior elements, and paraspinal soft tissues [6]. Neurologic involvement associated with SD in children ranges from no involvement to radicular pressure or intraspinal pressure, depending on the extent of the inflammatory process that evolves from the intervertebral disk. The epidural abscess is one of the most common and serious neurologic complications, affecting 4–38% of cases of SD [7]. The evacuation of the abscess is one of the few clinical conditions that require urgent spine surgery; however, if the abscess is limited and responsive to the antimicrobial treatment, the drainage procedure can be avoided. SD can lead to serious complications, such as spinal deformities and segmental instabilities, significantly reducing the quality of life for the affected patients [3,7]. When the infection spreads into the spinal channel, SD can cause catastrophic neurological complications, but early diagnosis and also a timely treatment might reduce these risks. However, unfortunately, these goals are difficult to achieve in children. The delay in diagnosis may lead to a worse outcome and lifelong disability [8]. Sapico and Montgomerie reported that 50% of patients had symptoms lasting for greater than three months before the diagnosis was established [9]. Diagnosis can be delayed because of the small number of cases of SD in children, together with the poor specificity of clinical manifestations and the inability of younger patients to verbalize the nature of the pain, especially in noncommunicating or uncooperative children. All these factors and the poor awareness of the disease may explain the difficulties in the diagnosis of pediatric SD. Moreover, identification of the causative infectious agents is frequently not performed, since the identification of the infecting organism requires invasive procedures, unless a blood culture is positive. For this reason, antimicrobial therapy targeted on drug susceptibility results of the causative infectious microorganism is rarely administered to the child and empirical therapy is generally prescribed. Another significant problem concerns the imaging, which is poorly contributive in the first period of the disease’s development.

Finally, the best therapeutic approach is not precisely defined—contrary to adult SD, for which treatment guidelines have been issued, no guidelines exist for pediatric patients [10]. Optimal antimicrobial treatment is still under debate in terms of molecule and duration. Several data suggest that a long-term therapeutic course is usually required to control symptoms and normalize laboratory tests [1,3].

## 2. Summary of Findings

### 2.1. The Included Studies

The selection process is shown in Figure 1.

Initially, 498 studies were retrieved. After title/abstract examination, 395 not pertinent articles were excluded. Therefore, 35 studies (12 case series including more than five children, and 23 case reports) were finally included in our literature review. Overall, 340 children were evaluated.

In the 12 case series studies, including 315 children, patients were retrospectively identified (1991–2019). Cohort size ranged between 6 [11] and 103 children [12], with seven studies including <20 children, and five studies including more than 20 children. Overall, 315 children were included.

The 23 case report studies included 25 children. Twenty-one studies reported one case, while only two studies comprehended two cases.

### 2.2. The Quality Assessment

Only two case reports [13,14] completely fulfilled the checklist criteria, scoring 8 out of 8 on the JBI Critical Appraisal Checklist for Case Reports.

In the remaining 21 articles, we found missing information or unclear information. Specifically, in 20 articles, the adverse events/the unanticipated events were not identified and described, while some articles did not report or reported partially data about the post intervention clinical condition (n = 3), about the interventions and treatment procedures (n = 3), about the patient’s history (n = 2), or about the patient’s demographic characteristics (n = 10). Moreover, two case reports did not provide a clear takeaway lesson.

Out of 12 analyzed case series, none of these completely fulfilled the checklist criteria—we considered not applicable the seventh question of JBI Critical Appraisal Checklist for Case Series. In all the 12 articles, we found missing information in one or more of the investigated aspects. The 10th question, about statistical analysis, was appropriate for only two study [12,15]. In four studies, inclusion criteria were not clearly described. In five studies, demographic data of the participants were not reported or reported partially. In eight studies, patient’s clinical information was not reported or reported partially. Outcomes and the follow-up results were missing in four articles. The quality assessments of the studies are reported in Figure 2 and Figure 3.

### 2.3. Demographic Characteristics 

Regarding the age range, it was extremely variable, ranging from 45 days [13] to 17.2 years [15]. The most frequently affected were children aged between 6 months and 4 years [1,11,16,17], even if in some studies, especially in the case reports, a later peak in the juvenile and adolescent group was reported. In only 6 studies, cases regarding children aged under 6 months were reported [5,8,12,13,15,18].

Chandrasesan et al. attempted to exemplify the age distribution, the related clinical features, and outcomes, and subdivided the children into those aged less than 24 months, and those aged 24 months or over [16].

Concerning the sex prevalence, 158/290 (54.48%) were the children in the male group, and 122/290 (42.06%) in the female one. One study did not report the sex of the 50 children [4].

### 2.4. Clinical Features, Physical Examination, and Comorbidities 

The most affected site was the lumbar spine, from L1–L2 to L5–S1, and the most affected segment was L4–L5, followed by L5–S1 [12,16,19]. The cervical spine involvement was mentioned in seven cases. According to these studies, it was clear that clinical features were different for age groups.

The percentage was calculated on a total of 237 children, since for the 103 children described by Dayer et al. [12], clinical features, except for neurological signs and fever, were not reported.

In the older ones (for Fernandez et al. children over three years [4], while for other authors over one year), the most commonly reported symptom was back pain, pre-existent and localized in the part of the spine affected by the infection. It was present in 90/237 (37.97%) children, and accompanied by limping, stiffness, and reduced movements. Spencer et al. reported that no child of their cohort size (n = 12) under 28 months complained of back pain, whereas all the children aged over 28 months had back pain to a varying degree [20].

In the toddlers, the most frequently reported symptoms were irritability/general malaise in 33/237 (13.92%) children and limping + refusal/inability to walk/to stand/to sit/to bear weight/to crawl in 118/237 (49.79%) children.

At the physical examination, the most frequent signs were antalgic positions/gait in 16/237 (6.75%) children, limitation of spinal movements (passive or active) in 38/237 (16.03%) children, and pain/tenderness at percussion/palpation in 21/237 (13.08%) children. In 32/237 (13.5%) children, there were also extra-skeletal manifestations, such as abdominal pain, constipation, and incontinence. Finally, neurological signs were seen in 13/340 (3.8%) children. Fever at the admission was present in 113/340 (33.23%) children.

The patients were almost all previously healthy children (PHC). Twelve children over 237 (5.06%) presented with comorbidities: anorexia, pre-existent spine deformities (isthmic spondylolisthesis and kyphotic deformity), Cushing disease, juvenile arthritis rheumatoid, hematological tumors, preterm babies, Wiskott-Aldrich disease, and sickle cell disease (SCD).

A total of 31/237 (13.08%) children suffered from previous infections (upper/lower respiratory tract infections, diarrhea).

Two children developed SD as a consequence of sepsis [8,13]. Three children developed SD after therapeutic procedures (lumbar puncture, dry cupping, and teeth extraction) [21,22,23]. Three children developed SD after the ingestion of a button battery [24,25]. Finally, 15/237 (6.32%) had previous trauma.

### 2.5. Laboratory Investigations

Regarding the laboratory data, the results from all the studies were heterogeneous and could not be merged analytically. Moreover, in some of these studies, laboratory data were not reported, or reported partially.

In the largest (103 children) multicenter study [12], the following mean values were reported: White blood cell count (WBC): 11,020/mm^3;C-reactive protein (CRP): 19.1 mg/L;Erythrocyte sedimentation rate (ESR) (performed in 98/103): 43.1 mm/h and in 80/98 >20 mm/h.

The blood cultures were performed in 299/340 (87.94%) children, but they were positive in only 40/299 (13.37%), showing poor sensitivity. In Dayer et al.’s study, 94/103 blood cultures were performed, but only 8/94 (8.51%) were positive [12].

The biopsy/needle aspirations technique, both CT-guided (computed tomography-guided) and intraoperative, was performed in 95/340 (27.94%) children and 44/95 (46.31%) yielded positive results.

The use of PCR was limited; in Dayer et al.’s study 13/103 children underwent PCR assay in the peripheral blood, and 4/13 (30.76%) were positive. However, PCR was also performed in the throat swabs of the same 13 children, they all yielded positive results. Finally, PCR was performed in the biopsy’s specimen, 8/13 (61.53%) were positive. The identified microorganism was always *Kingella kingae* (*K. kingae)* [12].

### 2.6. Microbiological Results 

As regards the etiology, the retrieved studies reported that in most cases SD derived from hematogenous spread, except for three case reports (0.88%), in which SD probably derived from infections of contiguous tissues or direct external inoculation from a diagnostic/therapeutic procedure (i.e., lumbar puncture).

The most frequently identified responsible pathogens were: *Staphylococcus aureus (S. aureus)* (n = 33), *K. kingae* (n = 18), Gram-positive cocci (n = 13), Gram-negative (n = 13), *Mycobacterium tuberculosis (M. tuberculosis)* (n = 11), anaerobic bacteria (n = 3), and one case of *Aspergillus* spp. Among the reviewed studies, three cases with more than one pathogen were registered. The involved pathogens were *Morganella morganii* + *E. coli*, *Peptostreptococcus magnus* + Gram positive cocci in chains, and finally *Fusobacterium nucleatum* + Gram negative bacilli [26]. 

They were detected by blood cultures, biopsies, and PCR. More precisely, excluding Dayer’s multicenter study, in which a clear distinction between the various pathogen research methods was not made, in 34 cases the responsible pathogen was identified by blood cultures, in 29 cases by both intraoperative and CT-guided biopsy, and in two cases by PCR. In 254 cases the blood cultures, intended as the first level in the research for the pathogen, were negative. Consequently, in 70/254 cases a biopsy was performed.

All three detection methods (blood cultures, biopsies, and PCR) were required in just two studies [12,15].

It is interesting to remark that Dayer et al. pointed out that patients with *S. aureus* infection belonged to the juvenile and adolescent age group, whereas *K. kingae* was always identified in children aged under four years [12].

### 2.7. Imaging Studies 

Regarding the imaging, both radiological findings at the admission and radiological control findings at the discharge from the hospital or during the follow-up were evaluated in 33/35 studies, including both case series and case reports.

Overall, imaging investigations were performed in 236 children. At the admission and during the process of diagnosis, radiography (x-rays) was performed in 173/236 (73.30%) children, and in 122/173 (70.52%) abnormalities were found. The most commonly reported abnormality was the erosions of the endplates, adjacent to the inflammatory process, followed by a decrease of the intervertebral disk height and the narrowing of the disk space. In a minority of cases, spinal deformities, such as a decrease of lumbar lordosis or the presence of a dorsal kyphosis, were observed.

Shah et al. described a case of tubercular SD (TSD); in that case, the X-rays demonstrated complete destruction of the vertebral body [14].

In Fernandez et al.’s study, 13/14 children with a diagnosis of vertebral osteomyelitis underwent X-rays. In total, 7/13 presented no abnormalities, thus, the X-rays were negative [4].

Magnetic resonance imaging (MRI) was mainly used to confirm the results of the X-rays and therefore confirm the diagnosis of SD. It was performed in 146/236 children (61.86%)—it was always diagnostic. The observed abnormalities were related to the vertebral bone involvement, the pathological enhancement of the disc signal in T1 and T2weighted sequences, and the tumefaction of the surrounding soft tissues. In 26/146 (17.80%) cases, the presence of an abscess/phlegmon was reported. Twenty out of twenty-six were paravertebral/muscle (psoas) abscesses and the others were epidural collections.

In Moreno et al.’s study, the use of MRI demonstrated the presence of nerve root damages in five children [27]. Among them, only two were accompanied by radicular symptoms: a case of hypotonia in the gluteus and lower extremities (L5 compression) and another with abolished osteotendinous reflexes and straight leg positive test.

Computed tomography (CT) was used to study the size and the entity of the bony defects. It was performed just in 20/236 (8.47%) cases; however, it was diagnostic in 18/20 (90%). The detected abnormalities were the same as MRI.

Bone scan (BS) was used in 88/236 (37.28%) children and it was diagnostic in 78/88 (88.63%). The radioisotope was Tc99; in only one case combined BS with Tc99 and gallium was used. The increased uptake of the radioisotope in the adjacent elements of the affected disk demonstrated the inflammatory process.

Single photon emission computed tomography (SPECT) was performed only in Barkai et al.’s study [19].

Finally, positron emission tomography (PET) was used in one child, in order to differentiate SD from other diseases, such as osteosarcoma of the vertebral body [28].

The available data of the follow-up imaging findings were various and heterogenous and in some studies they were not reported. Moreover, some authors did not report the precise imaging method that was used during the follow-up, even if an accurate description of the radiological findings after the clinical resolution of the SD would be useful.

We have noticed that in most cases, despite the fact that the child was healed from a clinical point of view with favorable outcomes, radiological sequelae were still evident at the last radiological check. Despite the resolution of the inflammatory process and the presence of a bony healing process, radiological residual abnormalities, and persisting degenerative changes were also commonly reported, such as fusion of the vertebrae, persisting narrowing of the disk space, loss of intervertebral disk height, and spinal deformities (especially kyphosis) [16,18,19].

### 2.8. Treatments 

The therapies, both intravenous and oral, were different and variable. In some studies, the therapy was not reported and in others, even if reported, it was incomplete.

The most used antibiotics were beta-lactams—both cephalosporins of the third generation, especially ceftriaxone (mentioned five times) and penicillin (particularly semisynthetic) [8,16,17]. Vancomycin and clindamycin were frequently used too. In 18 out of 32 studies, the authors reported double or multiple antibiotic therapy, while in the remaining 14 studies the chosen therapy was based on the use of a single drug.

The intravenous therapy (IVT) duration range was 1 week–25 weeks. The oral therapy (OT) duration range was 5 days–36 months.

The conservative therapy comprehended also the immobilization—19 studies reported the type of body immobilization. In most studies the authors reported the immobilization as part of the conservative treatment, mentioning body plaster cast/spinal bracing/corset therapy without specifying the type of orthosis. Only in four cases the authors did specify the type of orthosis, mentioning thoracic lumbar sacral orthosis (TLSO), hip spica, lumbar corset, and cervical collar. The duration of the immobilization, not always reported, was different among the studies, ranging from one month to two years.

On the other hand, surgery was used in 20 children out of 340 (5.88%) [8,15,17,18,27,29,30]. The applied surgery techniques were laminectomy, discectomy, and drainage of abscesses.

### 2.9. Follow up Results (Outcomes and Sequelae)

The outcomes, when reported, were in most cases favorable.

A large number of children were asymptomatic at the discharge, free of pain and with normal spinal movements.

Concerning the sequelae, the most frequently reported were radiological sequelae, especially residual kyphosis. In 29 cases, the maintenance of mild back pain/discomfort, restricted spinal movement with spinal deformities were reported, while in only one case recurrence of infection was documented. 

In Kayser et al.’s study, during the follow-up, 16/25 (64%) children were asymptomatic and free from pain. Four of twenty-five children (16%) presented restricted movement of the spine with reported local kyphosis, and 9/25 children (36%) complained of intermittent back pain [5].

These results are reported in Appendix A.

## 3. Discussion

In the present systematic review, 35 studies (12 case series and 23 case reports) were retrieved, involving overall 340 children with SD. Very few studies were published about this topic, and the literature lacks systematic knowledge of pediatric SD.

The purpose of this study was to analyze notions concerning pediatric SD, both pyogenic and granulomatous, also in an attempt to identify aspects of this disease that should be further investigated.

Regarding the age range of children in the included studies, it was extremely variable—children from six months to four years have been confirmed as the most frequently affected [1,11,16,17] despite the fact that a late peak in the juvenile/adolescent group was observed [15]. In six studies, cases of SD in neonates/infants under six months were reported [5,8,12,13,15,18].

The predominantly affected sex was not reported in the literature; however, in this systematic review, 54.48% were males and 42.06% were females. Thus, males emerged being more affected than females [12].

Interestingly, clinical features were different for age classes: in older children (>3 years), back pain was the main reported symptom—it was reported in 37.97% of children. In younger children (<3 years), the most commonly reported symptoms were irritability/general malaise, present in 13.92% of children, and limping and refusal/inability to walk, documented in almost half of the cases [4,20].

Nonspecific presentation in noncommunicating children led to a delay in diagnosis [16].

Fever at the admission was present in about a third of cases [11,17].

Almost all children were PHC, 5.06% of them had comorbidities [21].

Previous infections (viral or bacterial), present in 13.08% of cases, and previous traumas, in 6.32% of cases, were the most relevant problems reported in recent medical history [13,16].

The lumbar/lumbosacral spine was the most common area of involvement [12,16,19]. The underlying reason for susceptibility of the lumbar region to SD is not fully understood, but is likely to be partly related to venous drainage, through Batson plexus [26]. Conversely, the cervical spine involvement was mentioned in seven cases.

In addition to laboratory tests, the diagnosis was obtained with blood cultures, which were performed in 87.94% of children but were positive in 13.37%, and biopsies, performed in 27.94% of cases but positive in almost half of these.

The most commonly isolated pathogen was *S. aureus* (n = 33)*;* however, many authors reported an increasing presence of *K. kingae* (n = 18)*,* often isolated with PCR techniques [12]. These microbiology findings were consistent with the published literature.

Concerning imaging investigations, both radiological findings at the admission and radiological control findings at the discharge from the hospital or during the follow-up, were evaluated. MRI was the most reliable method in detecting SD at the admission, with high sensitivity and specificity [4]. It was used to confirm X-rays results, and it was always diagnostic. Advantages of MRI, in addition to the lack of ionizing radiation, include multiplanar capability, superior soft-tissue contrast, and evaluation of the neural structures. It also allows the detection of other conditions that may require surgical intervention, such as soft-tissue abscess, bone destruction, and involvement of neurologic components.

CT was used in a minority of cases. It was mainly used to study the size of the defect, following radiography, especially in those cases of suspected neurological damage, with neurologic findings at physical examination. In older studies, it was used instead of MRI, often in conjunction with BS. CT is considered the best modality at delineating bony abnormalities, including early endplate destruction, loss of disc height, and erosive changes. Epidural extension of infection may also be identified. CT can be particularly useful in patients for whom MRI is contraindicated because of implanted devices, or if MRI is unavailable. However, CT is inferior to MRI in the visualization of neural tissue and abscesses. Moreover, the use of ionizing radiation should be avoided in paediatric patients, obviously due to the young age and the possible related damages. BS was also reported, especially in older studies. During the follow up, residual radiological abnormalities were found [16,18,19].

The conservative treatment consisted of bed rest, immobilization, and antibiotic therapy. Unfortunately, there are no guidelines for antibiotic therapy; therefore, antimicrobial treatments (IVT and OT) were various and heterogenous, both regarding the choice of the molecule and the duration. In many cases, treatments were not even reported or reported partially. Thus, it is difficult to standardize the course and content of therapy in pediatric SD. However, the most used antibiotics were beta-lactams, mainly third-generation cephalosporins, oxacillin, and flucloxacillin [8,16,17]. Nineteen studies reported the type of body immobilization and the duration too.

Surgery was used in 5.88% of all children [8,15,17,18,27,29,30]. The decision to perform surgery was based on the need of spine stabilization in patients with a consistent risk of progression of severe kyphosis, the need for drainage of large or multiple abscesses after antibiotic therapy’s failure, clinical and imaging findings of bone deterioration with extensive vertebral body destruction, and neurologic deficits or motor regression compatible with lower motor neuron injury. The use of surgery in these few and selected cases improved the patient’s outcome and was fundamental in those conditions in which the conservative treatment had failed.

A favorable and relatively benign long-term outcome was observed in the majority of cases; however, some studies have demonstrated sequelae such as recurrent mild back pain and spinal deformities in children [15,16,26].

Through the evaluation of major historical series and their comparison with the most recent ones, it is possible to comprehend how the medical approach to SD has changed over time. As regards the pathogenesis of the disease, it is interesting to consider Menelaus et al.’s study (1964). They included 35 children with SD, with back pain at physical examination and elevated ESR at laboratory investigations. The authors reported how their symptoms subsided after immobilization without antibiotics, therefore they suggested a viral etiology of SD and advised the use of antibiotic therapy only for febrile children [31]. Otherwise, Wegner et al., in 1978, describing a cohort of 41 children, presumed a bacterial etiology of the disease, and in more recent studies the bacterial nature of the disease has been ascertained [3,32].

For what concerns the clinical manifestations, they appear related to patient’s age. Crawford et al., in 1991, grouped patients by symptoms and age in three classes, as suggested also by Wegner et al. The first group consisted of three years old patients and younger, among these children, the most common clinical manifestations were: straight leg positive test, refusal to walk (70%), localized tenderness in the spinal region (50%), and abdominal pain (30%). The second group consisted of children between the age of three and nine years, for whom the most common clinical features were: back pain (30%), refusal to walk (75%), and tenderness in the spinal region (50%). The last group consisted of children older than nine years and back pain was the leading symptom [33]. In 2018 the results of the multicenter retrospective study by Dayer et al. suggested that SD has a predominantly biphasic distribution, rather than the historical triphasic distribution, with only exceptional infections (1%) in children being under the age of six months, a higher incidence (79%) in early childhood, between the age of six months and four years, and a later peak (20%) in the juvenile’s and adolescents’ group [12]. Clinical manifestations of SD can significantly vary, though delay in the diagnosis is common. Garron et al. in 2002, in their study of 42 children, described an average interval time from onset symptoms to diagnosis of 42 days, underling how the initial clinical presentation was often misleading [34]. Likewise, Bonfiglio et al. in 1973 defined the onset symptoms insidious and underlined how the average time from first symptoms to the day of admission to the hospital was usually more than three months [35]. 

Regarding the imaging findings, Bonfiglio et al. accurately described the evolution of radiological features of a six years old boy with SD. The X-rays at admission appeared normal. Two weeks later, X-rays showed narrowing of disk space with irregularity of the distal portion of the vertebral body. At three months, the narrowing was more evident. At two years, X-rays showed partial restoration of the disk. Nine years after the onset, at age 15, in addition to the persistence of narrowing of the intervertebral disk space, the vertebral bodies showed an enlargement [35]. Recent studies have described how the use of X-rays, although helpful in advanced cases, is otherwise limited for diagnosis. Radiographs can take up to six weeks to show changes and modification of the vertebral bodies, therefore MRI has replaced X-rays as modality of choice for diagnosis of SD, with high specificity and sensitivity [3,4].

SD is a rare disease in children; however, if not treated promptly, it can lead to significant sequelae. Therefore, a high index of suspicion, an early diagnosis, and a targeted treatment are necessary to reduce these risks. SD should be suspected in children with back pain, antalgic gait, and refusal to walk, but also with unspecific symptoms such as general malaise and irritability, especially in younger children.

Imaging plays a very important role for the diagnosis of SD. MRI represents the gold standard in the investigation of infections of the spine, particularly in the early stages of the disease when these changes are not apparent in other imaging tests, including CT and X-rays. This may help reducing diagnostic delay, while also guiding the extent and duration of treatment.

Medical management with antibiotic therapy is the treatment of choice in SD, however the best therapeutic approach is not precisely defined—contrary to adult SD where treatment guidelines have been issued, no guidelines exist for pediatric patients. The problem mainly concerns the duration of the antibiotic therapy. In literature, some authors have suggested a short parenteral therapy, as suggested for other types of osteomyelitis. However, the Infectious Diseases Society of America (IDSA) recommends a total duration of six weeks of parenteral therapy [10]. Due to these uncertainties, we believe that specific guidelines for the management of SD in the pediatric age would be desirable.

However, this systematic review has some limitations. First of all, the small number of retrieved studies, their retrospective nature, and their substantial methodological differences. The difference between them also includes different geographical origins of the analyzed patients, the years in which they were written, and different the cohort sizes. Moreover, some of these studies reported incomplete data, especially regarding antibiotic therapy.

## 4. Search Strategy and Quality Assessment

### 4.1. Study Design and Search Strategy

In order to evaluate current literature evidence regarding the management of pediatric SD, we performed a systematic review of the literature, according to the Preferred Reporting Items for Systematic Reviews and Meta-analyses (PRISMA) guideline recommendations [36] (Appendix A).

On 1 July 2020, the MEDLINE database (Medical Literature Analysis and Retrieval System Online; through the PubMed interface) and the EMBASE database (Excerpta Medica database) were searched, using the following PubMed search strategy: (“discitis”[MeSH Terms] OR “discitis”[All Fields] OR “spondylodiscitis”[All Fields]) AND (“child”[MeSH Terms] OR “child”[All Fields] OR “children”[All Fields] OR “child s”[All Fields] OR “children s”[All Fields] OR “children”[All Fields] OR “child”[All Fields]) AND (“pediatrics”[All Fields] OR “pediatrics”[MeSH Terms] OR “pediatrics”[All Fields] OR “pediatric”[All Fields] OR “pediatric”[All Fields]). Moreover, all reference lists from the retrieved articles were reviewed to identify any additional pertinent articles. No language restriction or publication date restrictions were applied. Comments and letters to editor were excluded. All retrieved articles were reviewed by two observers (EC and IF). In the review of the articles identified, particular attention was paid to the study design, its methodological characteristics, clinical features, radiological findings and therapeutic strategies.

### 4.2. Data Extraction 

Data were collected and entered into an electronic database (Microsoft Corporation, 2018 Microsoft Excel, Redmond, WA, USA).

For each study the following information was retrieved and summarized in Appendix A: author, year of publication, country, study design, number of children, age, sex, ethnicity, clinical features, comorbidity, laboratory data, blood culture or other microbiological investigations’ results, biopsy/needle aspiration results, polymerase chain reaction (PCR), identified pathogen, affected spinal segment part, antibiotic therapy (intravenous and oral), immobilization, surgery, sequela, and outcomes.

We also collected data regarding the radiological findings, both at the admission and after the discharge or during the follow-up, and summarize findings in Appendix A.

### 4.3. Study Eligibility and Quality Assessment 

The articles included provide epidemiological, clinical, diagnostic, and/or therapeutic data about children with SD, both pyogenic and granulomatous.

Firstly, we screened titles and abstracts to discover eligible studies, and then we analyzed all full texts for the final evaluation. Two investigators (EC and IF) independently reviewed and evaluated every study. We considered a study eligible when the following criteria were met: (1) population were children; (2) a diagnosis of SD was made with specified diagnostic criteria; and (3) epidemiological, clinical, diagnostic, management, and outcome data were reported. Exclusion criteria were: (1) not relevant topic (not appropriate population or not appropriate outcome); (2) nonoriginal studies (e.g., literature reviews, guidelines, duplicate articles, comments); and (3) studies not reporting useful clinical data about patients.

The quality of the eligible studies was evaluated by two authors independently (EC and IF), using different methods according to the study design: The Joanna Briggs Institute (JBI) Critical Appraisal Checklist for Case Reports [37] and the Joanna Briggs Institute (JBI) Critical Appraisal Checklist for Case Series [38].

### 4.4. Statistical Analysis 

Statistical analysis was performed using Microsoft Excel (Microsoft Corporation, 2018). Categorical variables were expressed as the number of cases (N) and percentages (%).

## 5. Conclusions 

In conclusion, this review provides the main epidemiologic, etiologic, clinic, and diagnostic features of pediatric SD, both pyogenic and granulomatous, as well as an up-to-date overview on its management and outcomes. However, our paper emphasizes the significant gaps of the literature regarding this important topic.

## Figures and Tables

**Figure 1 antibiotics-10-00030-f001:**
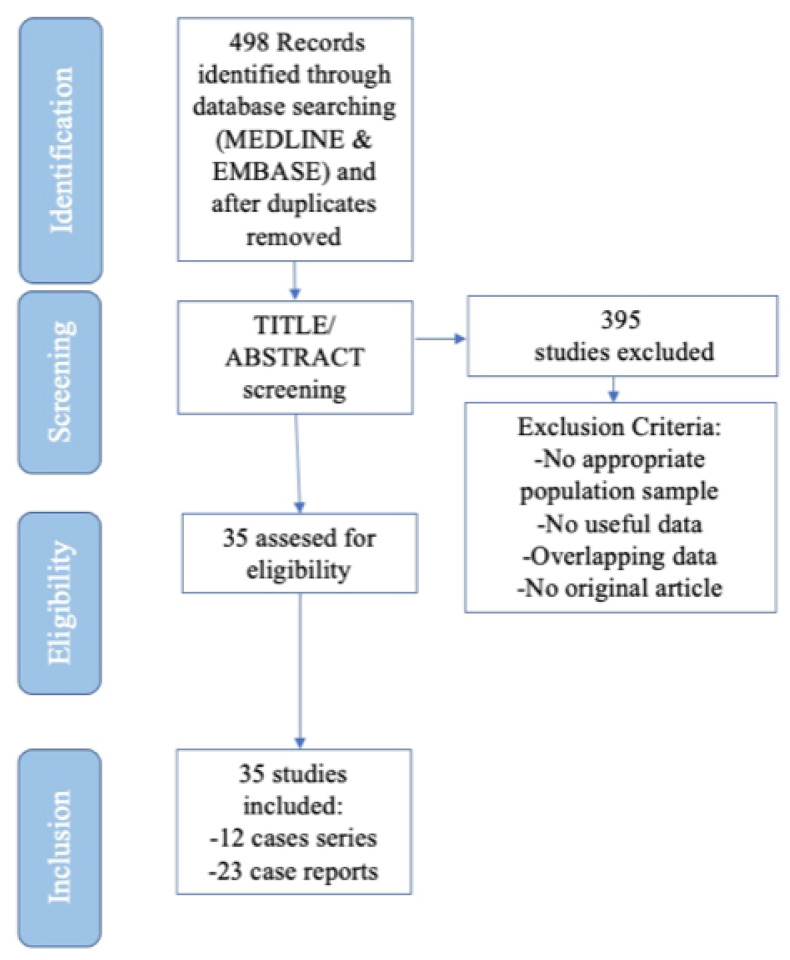
Flow chart highlighting the process of study selection for each step of the Systematic Review, following Preferred Reporting Items for Systematic Reviews and Meta-Analyses (PRISMA) guidelines.

**Figure 2 antibiotics-10-00030-f002:**
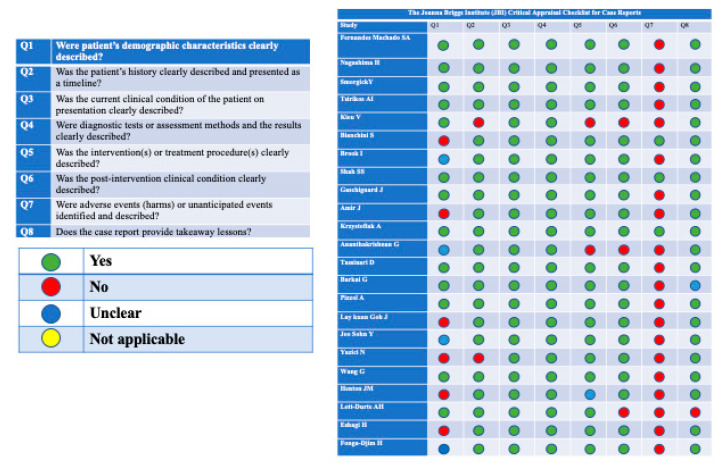
Quality assessment of the included studies. The Joanna Briggs Institute (JBI) Critical Appraisal Checklist for Case Reports.

**Figure 3 antibiotics-10-00030-f003:**
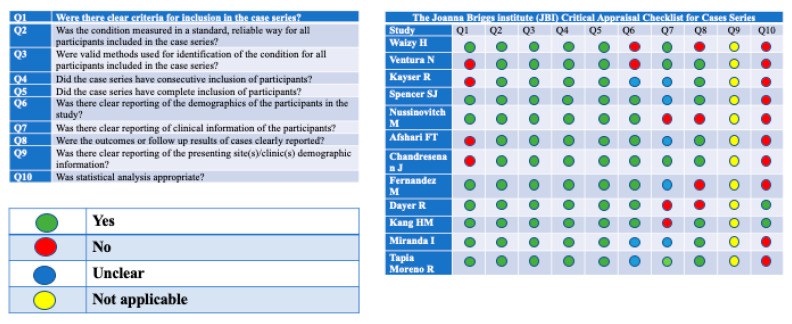
Quality assessment of the included studies. The Joanna Briggs Institute (JBI) Critical Appraisal Checklist for Case Series.

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
