# Peer review of "Characteristics, Management and Outcomes of Spondylodiscitis in Children: A Systematic Review"

_antibiotics, 2020, doi:10.3390/antibiotics10010030_

Round 1

Reviewer 1 Report

This is an interesting review on a subject that is in dire need of good information. I commend the authors for addressing this topic but I believe they could convey even more interesting information to the reader, such as the etiology of SD, indications for CT scan vs MRI, indication for surgery, etc.

I believe that the sections that are most interesting to the reader, such as treatment and microbiological results are far too short and should be extended with results. This review does not include even one table, which would be helpful to compare the results of different studies in these categories, respectively. Please try to assess and interpret the studies' finding to formulate recommendations for the reader.

Introduction

You state "two aspects of the same spectrum, which 34 appear in two different stages of the pathological process": please detail the two different stages and why they are now considered part of the same pathological process in comparison to other studies.

Authors should mention the incidence rate of epidural abscess and associated  neurologic impairment and underline that this is one of the few indications for urgent spine (decompression) surgery.

Results

Please also the ethiology (haematogenous? per continiuitatem?) with their respective incidence rate, as well the associated co-morbidites for the SD.

"use of MRI demonstrated the presence of nerve roots damages in 5 184 children" nerve root damage is difficult to assess in MRI and is usually assessed clinically first (dysesthesia? muscle weakness? etc.). Please detail if clinical findings corroborated the injuries seen in imaging.

"19 studies reported the type 215 of body immobilization and the duration." Please detail a short/summarized list the types of immobilization and duration that were employed.

The section "Microbiological Results" is very brief and needs to be extended. How many infections with more than one pathogen were registered? Please detail how these pathogens were found, what is the rate of detection/sensitivity/specificity for the blood cultures, biopsies, and PCR? How many studies required all of them? How many cases reported negative microbiological tests and required further investigation (and detail what was done instead)? There is plenty of information that could be detailed here.

Likewise for the "Treatments" section, which is far too short. Please report the rate of single and/or double/multiple antibiotic therapy and if this implied a variation in the duration of antibiotic administration.

Discussion

please detail why CT was used in those few cases and under which justification if MRI is the predominant diagnostic method. How did they justify the radiation exposure in children, with which intention did they indicate a CT scan? Was it done additionally to MRI imaging?

When was surgery indicated and for what reason? How was it justified (risk of deformity, extensive destruction, etc.)? Did it improve the outcome compared to conservative treatment?

Reviewer 2 Report

Thank you very much to give me the opportunity to read this systematic review about characteristics, management and outcomes of spondylodiscitis in children.

The different aspects of the review are well analysed. It is, to my opinion, an article whose findings are important not only to those with closely related research interests, but to every paediatrician.

However, there are several points in this review, which deserve a reflection or even a modification.

  1. Most of the time, studies about spinal infections focus about non tuberculous spondylodiscitis. Tuberculous spondylodiscitis are rare entities in our latitudes, with specific evolution and they should not be included in this study, to my opinion.
  2. Analogously, only primary pyogenic spondylodiscitis should be taken under consideration; in fact, children who developed spondylodiscitis after lumbar puncture or after ingestion of a button battery should be removed from the study to my opinion.
  3. I am not convinced that isthmic spondylolisthesis or kyphotic deformity may be considered as spine damages…
  4. Line 154; throat swabs instead of swans
  5. It will interesting to have the mean age for children with positive cultures due to S aureus, and the mean age for children with positive cultures due to K. kingae.
  6. I am especially very surprised that you excluded major historical series from the study, such as Crawford et al 1991, Wenger et al 1978, Menelaus et al 1964, Garron et al 2002, Bonfiglio et al 2006, Digby et al 1979; …..)

Round 2

Reviewer 1 Report

The authors have correctly addressed all issues raised. I recommend this paper for publication.

Merry X-mas to everyone!